# Epistemic trust towards teacher questionnaire: Development and preliminary validation

**Alex Desatnik**[1,2,3,4]*, **Maxim Yakubovskiy**[4,5]*, **Sergei Tarasov**[6], **Peter Fonagy**[1,2]

**1** Research Department of Clinical, Educational and Health Psychology, University College London, London, United Kingdom, **2** Anna Freud National Centre for Children and Families, London, United Kingdom, **3** Open Door Young People Service, London, United Kingdom, **4** London Gates Education Group, London, United Kingdom, **5** Department of Educational Policy Studies, School of Education, University of Wisconsin-Madison, Madison, Wisconsin, United States of America, **6** Institute of Education, National Research University Higher School of Economics, Moscow, Russia

* alex.des@mail.com (AD), yakubovskiy@wisc.edu (MY)

## Abstract

Epistemic trust (ET) is a well-established concept in evolutionary developmental psychology. It refers to the trust recipients place in their informants, which in turn makes them more attentive, thoughtful, and engaged in processing and acquiring information. This concept plays a key role in educational contexts, as leveraging this innate mechanism could help teachers to improve learning outcomes, including those beyond the classroom. This paper describes the development and validation of a new self-report measure, the Epistemic Trust Towards Teacher (ETT) scale, designed to assess the unique role of ET in education. The study focuses on the development of the scale, its factor structure, and construct validity. The study participants were 224 middle and high school students aged 11–15 (58.5% female). Confirmatory factor analysis supported a three-factor structure with good model fit ($\chi^2$/df = 1.9; CFI = 0.970; TLI = 0.965; RMSEA = 0.062), consisting of Trust (ETT_T), Mistrust (ETT_M), and Generalisation (ETT_G) subscales. Reliability was acceptable to excellent across the three subscales (Cronbach's $\alpha$ = .75–.88). The factors showed good discriminant validity and aligned with theoretical expectations. The ETT scales correlated in theoretically consistent ways with established measures of teacher-student relationship quality, working alliance, non-verbal immediacy, and student motivation. Overall, this study provides preliminary support for the ETT as a relevant and psychometrically sound instrument in the field of educational psychology. Further research is required to investigate the ETT's performance across a wider age range and diverse educational settings.

## Introduction

Epistemic trust is defined as 'trust in the authenticity and personal relevance of interpersonally transmitted knowledge' [1]. The transmission of knowledge, or the

**Data availability statement:** Our study's minimal underlying data are within the manuscript, and the data are now available on Figshare. DOI: 10.6084/m9.figshare.29649758.

**Funding:** The author(s) received no specific funding for this work.

**Competing interests:** The authors have declared that no competing interests exist.

paradigmatic process of teaching and learning, is believed to be an evolutionary construct that distinguishes human beings from other species [2], making us a knowledge-seeking species [3]. The exploration of the humans' instinct to learn [4,5] and its implications lies within the field of natural pedagogy, where the development of Tomasello's ideas culminates in the assertion that teaching and learning in this regard are inseparable and constitute an essential paradigm in a human life [6] and the transmission of culture and knowledge [7]. It is posited that if the instincts to teach and learn are innate, there must also exist natural mechanisms that facilitate learning and teaching — termed 'keys' to the 'gates' of effective learning. In this context, epistemic trust (ET) is proposed as evolutionary protected channel that enables the sustainable acquisition of knowledge, a crucial characteristic of which is its generalization across multiple contexts [6,8]. ET is thought to be established through ostensive cues (OCs) [6,9], which are communicational "signals" conveyed by the individual imparting the information to the recipient. A common feature of these cues is that they demonstrate to the recipient that they are recognized as an individual agent (e.g., name-calling, smiles, physical proximity, etc.) [10]. In contrast, epistemic mistrust, which entails assuming malevolent intentions based on the recipient's interpretation of the communicator's signals [11], is linked to failures in social learning and an exaggerated manifestation of epistemic vigilance. This vigilance, which refers to humans' innate mechanisms that assess the believability of the incoming information and the credibility of its source [12].

ET is considered to be a critical factor in the resilience towards and risk of psychopathology and is believed to be associated with and facilitated by attachment and mentalizing [1,7,13]. Furthermore, ET is also thought to be central to one's ability to benefit from psychotherapy and to form a working therapeutic relationship (alliance) [2]. In addition to playing a pivotal role in the development of epistemic trust, these two concepts are both known to contribute to one's capacity to learn and achieve academically [14–18].

While most current studies of ET originated from the field of developmental psychopathology [2,7,19], it can be argued that, as an evolutionary mechanism underpinning the transmission of knowledge in humans, epistemic trust is of paramount importance in the educational process. Traditional education encompasses the essential paradigmatic two-way "human" activity of teaching and learning, which necessitates establishing a relationship. This human-to-human interaction, which fosters a trusted relationship, is critical from childhood trough young adulthood [20].

However, since not all communicators are equally successful in facilitating ET [6], it appears essential that this skill be considered a crucial component of teachers' competencies, warranting significant attention to its assessment and development.

Moreover, it can be argued that epistemic trust mitigates the risk of "pseudo-education" and surface learning [21,22]. As students cultivate a relationship of trust with their teacher, they are more likely to generalize the information obtained within the classroom and apply it in out-of-classroom (i.e., "real-world") contexts. It is widely accepted that the eventual generalization of the acquired knowledge across diverse contexts is a fundamental purpose of education and one of its most critical

educational outcomes. Consequently, education paradigms have been shifting to focus on the real-world applicable outcomes through student-driven, problem-solving, and design-based approaches [23–25], with some even describing this shift as a "world-centered" approach [26]. Therefore, ET holds significant interest for educational research focused on educational outcomes.

While it is assumed that ET is an essential component of education in general, at the same time, when applied to a classroom context, it is crucial to explore not only individual differences in students — such as their overall capacity to develop epistemic trust relationships — but also the variations in attitudes towards a particular teacher, academic achievement, motivation to study, and the ability to express and utilize their ET relationships, all of which may affect the development of epistemic trust.

More broadly, the importance of the quality of teacher-student relationships (TSR) for various factors related to educational outcomes is well established. Extensive research has been conducted into the impact of TSRs on academic achievement [27–30], student motivation [31–34], attitudes towards to learning and engagement [35–37], and the role of emotions [21].

Several self-report measures have been developed to assess constructs that are theoretically related to ET through the evaluation of teacher behaviors. These include among others measures aimed at assessing TSR quality, such as the Student-Teacher Relationship Scale (STRS) [38] and the Teacher-Student Relationship Inventory (TSRI) [39]. Other measures focus on observable physical behaviors of teachers, capturing aspects of non-verbal immediacy, such as appropriate or inappropriate physical proximity, smiling, and body posture (Nonverbal Immediacy Scale (NIS) [40]). While these constructs are partially overlap with ET, none of them seem to comprehensively capture all its facets of ET, particularly the generalization of knowledge acquired through the teacher-student relationship. To date, there are no validated measures allow for the exploration of epistemic trust between teachers and students.

Recently, a general measure of ET has been published [41]. However, this measure assesses only individuals' general tendency for ET and mistrust without considering context or relationships with specific individuals. Consequently, it is less relevant for the contextual evaluation of ET in education concerning a specific relationship with a particular teacher, which necessitates its own targeted assessment. Furthermore, this measure was developed for adults and therefore may be less suited for measuring ET in school-aged students.

Another important consideration is that, although ET is highly applicable to education, the concept originates from the field of developmental psychology and evolutionary psychology, where it is anticipated that aspects specific to the educational contexts may not be fully captured within the developmental or clinical frameworks. Therefore, it is necessary to develop an instrument that focuses on the particular teacher-student relationship and the development of ET within such a relationship. This approach would allow for examining the influence of ET on students' learning capabilities and the unique role of an individual teachers in facilitating this process. Such an instrument is essential for enabling larger-scale studies to investigate the hypothesized unique role of ET in education, its effect on motivation, and its impact on academic outcomes.

## The present study

This paper aims to outline the psychometric properties of a new self-report measure of Epistemic Trust Towards Teachers (ETT). It describes the development and initial validation of this measure, focusing on the ETT's development and factor structure within a sample of middle and high school students (n = 224). This study also examines the construct validity of the ETT by exploring its convergent, and divergent validity.

## Methods

### Development of the epistemic trust towards teachers questionnaire

**The challenges in developing an education-specific ET self-report measure.** ET is a multifaceted phenomenon comprising various elements, including, within the educational context, the actions of the teachers, the student's perception of those actions, and the resulting impact on learning. During instruction, the teacher provides multiple

ostensive cues (OCs) [6,9], which may or may not be effectively "received" by the student, allowing them to feel recognized as an agent. This interaction can establish epistemic trust if the student regards the teacher as trustworthy, or alternatively, mistrust if they do not. When ET is established, knowledge is then successfully transmitted through a prioritized channel, perceived as relevant and generalizable. Notably, many elements of ET are likely to develop outside of conscious awareness: many ostensive cues may be "sent" by the teachers without deliberate intent (implicitly), and students' interpretations of these cues are not always consciously determined.

The challenge in designing a measure of ET towards a specific teacher lies in attempting to capture a broad spectrum of interactions that pertain to epistemic trust or mistrust. It is important to not only to account for the presence or absence of OCs — a feature measurable through tools assessing non-verbal immediacy — but to ascertain whether the OCs were both sent and received, and whether the meanings intended by the teacher align with those perceived by the student. In other words, it is essential to determine if the ostensive cues opened a channel for effective communication and the retention of generalized information.

The ETT was designed as a student self-report questionnaire. Its development carefully considered that some teacher behaviors in classroom may not be intentional, and students' interpretations of these behaviors may not involve a conscious decision-making. Nonetheless, students can report on teachers' physical behaviors, such as aspects of non-verbal immediacy.

## Study hypothesis

We investigated the reliability and validity of the ETT subscales in a sample of middle and high school students (n = 224). Exploratory factor analysis (EFA) and confirmatory factor analysis (CFA) were employed to examine the factor structure. In line with theoretical formulations, we anticipated a three-factor structure, with scales assessing trust, generalization, and mistrust towards teachers, to provide the best model fit for the data.

The convergent and divergent validity of the ETT was evaluated by examining relationships between the ETT and demographic variables, as well as measures theoretically aligned with ET [38,41]. We expected no demographic associations with any of the ETT scales. We hypothesized that the scales indicating the presence of ET towards a teacher (Trust and Generalization) would correlate positively with the subscales assessing the positive aspects the working alliance (WAI) [42], non-verbal immediacy (NVI) [43], and the Teacher–Student Relationship Inventory (TSRI) [39], and conversely, show negative correlations with scales indicating adverse aspects of these measures For the Mistrust scale, we anticipated an opposite pattern of associations.

Additionally, we expected the ETT scales to show significant associations with student motivation, measured using the Student Motivation Scale (SMS)) [44]. Theoretically, ET, and particularly generalization, were anticipated to have a meaningful impact on student motivation and subsequent academic performance. However, we expected these correlations to be modest, as motivation is likely influenced by additional factors beyond the quality of trust towards the teacher.

## The development of the scale

Initially, 109 items were developed based on the existing ET questionnaires and an extensive literature review in order to capture high or low levels of factors associated with epistemic trust or mistrust as well as factors that were conceptually expected to influence ET. Participants' responses were rated on a 5-point Likert scale, anchored as "strongly disagree" (= 1), "somewhat disagree" (= 2), "neither agree nor disagree" (= 3), "somewhat agree" (= 4), and "strongly agree" (= 5). This approach allowed participants with high levels of trust to "strongly agree" (= 5) with certain statements, while those exhibiting high levels of mistrust would "strongly disagree" (= 1), with reverse scoring applied for negatively worded items.

The 109 items were categorized into six groups, designed to reflect the full scope of ET within an educational context:

1. *Physical ostensive cues (OCs)*: Observable, basic, and universal cues that are consistent with experimental evidence (e.g., "This teacher often smiles at me").

2. *Recognition of student as an agent*: Secondary level cues focusing on the extent to which the student feels acknowledged as an individual (e.g., "This teacher notices when I don't understand something").

3. *Humor:* Shared laughter is considered a strong cue, creating an experience of a shared mental space, along with high-level contingency, and mutual joy in communication (e.g., "Sometimes we laugh and joke together with this teacher").

4. *Trustworthiness:* A core facet of epistemic trust, presumed to be triggered by ostensive cues (e.g., "I do not trust this teacher").

5. *Teacher's perception of me*: This scale assesses how the student perceives the teacher's view of them, reflecting the effectiveness of OCs in that specific teacher-student relationship (e.g., "I feel that my success is important to this teacher").

6. *Generalization of knowledge*: Measures knowledge perceived as subjectively relevant and applicable across multiple contexts, aligning with the primary aim of education to foster deep, retained, and transferable learning (e.g., "Some things I learned from this teacher were useful outside the class").

Each category included both positive and negative items. For instance, in Physical ostensive cues, items included "This teacher often smiles at me" and "This teacher often does not say hello to me even when they see me." The initial 109 items then underwent review and rating by seven international experts in developmental psychology, educational research, and psychometrics. Each item was rated for clarity, relevance to epistemic trust, and potential redundancy. Items were retained if they were rated by at least five of the seven reviewers as clear, relevant, and non-redundant. Items lacking conceptual alignment with epistemic trust, judged as overly ambiguous, or considered repetitive were removed. While no formal inter-rater reliability statistics were computed, a structured Delphi-like process ensured broad consensus. Reviewers provided both quantitative ratings and qualitative comments, which were synthesized to guide the refinement of the item pool. This process resulted in a refined set of 52 items.

## Participants and procedures

The ETT was administered to 224 school students aged 11–15 from several schools in Moscow between 05/04/2021 and 05/06/2021. All students were recruited through their schools. Before the study, participants' parents or guardians received a consent form, had the opportunity to inquire about the research, and voluntarily consented on behalf of the students. Anonymous questionnaires were distributed by the researcher at the end of a lesson. The study was approved by the Higher School of Economics (HSE) Committee on Interuniversity Surveys and Ethical Assessment of Empirical Research and the London Gates Education Group Ethical Review Board (No 53485). Data was securely stored, with no third-party access, and was destroyed upon completion of the analysis.

The participants' characteristics are summarized in Table 1.

A dedicated team of four research assistants was assembled and trained to collect the data, ensuring thorough familiarity with the questionnaire, ethical guidelines, and data collection protocols. The researchers coordinated with school administrators to finalize logistics, including the selection of distribution locations and schedules. They also introduced the study to the participants, emphasizing the voluntary nature of their participation and providing clear instructions for completing of the questionnaires.

## Other measures

To assess the correlations with existing constructs and evaluate the validity of the newly developed questionnaire, the participants were also asked to complete several additional questionnaires.

**Table 1. Participants characteristics (N = 224).**

| | Mean (SD) or n (%) |
|---|---|
| Age (years) | 13.21 (1.43) |
| Gender | |
| female | 131 (58.5%) |
| male | 93 (42.5%) |
| Mother's education | |
| Higher education | 196 (87.5%) |
| Unfinished higher education | 6 (2.7%) |
| Further education | 3 (1.3%) |
| No info | 19 (8.5%) |
| Father's education | |
| Higher education | 188 (83.9%) |
| Unfinished higher education | 3 (1.3%) |
| Further education | 5 (2.3%) |
| No info | 28 (12.5%) |
| Gadgets at home | 11.61 (6.49) |
| Own room | 199 (88.8%) |
| Physical health (from 1 to 5 ) | 4.27 (0.79) |
| Familiar with the teacher | |
| less one year | 159 (71.0%) |
| one year | 46 (20.5%) |
| two years and more | 11 (4.9%) |
| No info | 8 (3.6%) |
| Do you like going to school? (from 1 to 5 ) | 3.83 (1.01) |
| How important are grades to you? (from 1 to 5 ) | 3.58 (1.04) |
| Do you like studying at your school? (from 1 to 5 ) | 4.52 (0.72) |
| Do you like studying at all? (from 1 to 5 ) | 3.94 (0.95) |

The first questionnaire was the 12-item Working Alliance Inventory (WAI) [42], which utilized a 5-point Likert scale and measured three separate factors — goal, task, and bond — as well as one overarching factor. Cronbach's α was = .83, .76, and .80 for each respective subscale and .91 for the common factor.

The second measure was the Non-Verbal Immediacy (NVI) [43] questionnaire, a 13-item instrument using a 7-point Likert scale to assess two factors: positive and negative immediacy behaviors. Cronbach's α for the subscales was .68 and .76, respectively.

The third questionnaire, the Teacher–Student Relationship Inventory (TSRI) [39], was a 14-item measure using a 5-point Likert scale, with three distinct factors — satisfaction, instrumental help, and conflict. Cronbach's α for each sub-scale was .90, .83, and .82, respectively.

The fourth questionnaire measured student motivation through the Student Motivation Scale (SMS) [44], a 12-item instrument using a 7-point Likert scale, with Cronbach's α reported at .89.

Additionally, student motivation was assessed using the final two questions from Table 1. These measures enabled comparisons between two facets of motivation: general motivation for studying and motivation specific to the educational center.

## Study

### Data analysis

In the first stage, an exploratory factor analysis (EFA) was conducted to determine the factor structure of the developed questionnaire. All the responses for the analysis were recoded in the direction of increasing epistemic trust, ensuring that higher scores corresponded to higher levels of ET. To establish the number of factors, the Cattell's scree test [45] and the Kaiser criterion [46] were utilized; however, the final decision on number of factors was based primarily on interpretability.

In the next stage, confirmatory factor analysis (CFA) was employed to select items for a shortened version of the questionnaire and to evaluate the fit of the derived model. Given the categorical nature of the data, the Weighted Least Squares Means and Variance adjusted (WLSMV) robust estimator was applied [47]. The model fit was assessed using the following indices: $\chi^2$/df ratio, comparative fit index (CFI), Tucker-Lewis index (TLI), the root mean square error of approximation (RMSEA), and the standardized root mean square residual (SRMR). An acceptable model was defined as one where $\chi^2$/df < 3, CFI and TLI > .95, RMSEA < .06, and SRMR < .08.

At the final stage, convergent and divergent validation was conducted by examining the relationships between the ETT subscales and other measures related to epistemic trust using Pearson correlation.

## Results

### Exploratory factor analysis

Cattell's scree test suggested a three-factor solution, as the last significant decline in eigenvalues occurred at the third point (Fig 1).

According to the Kaiser criterion, a four-factor solution would be appropriate, as the eigenvalues from the fifth factor onward are below 1. However, solutions with more than three factors were challenging to interpret due to the limited number of items loading onto the fourth and fifth factors. Thus, a three-factor solution was selected based on interpretability. The Varimax- and oblimin-rotated solutions for the three factors were similar. The first three factors accounted for 39.8% of the variance, with the results presented in Table 2. Items that were rescored are marked with an asterisk (*). The "uniqueness" values in the table indicate how much of the item variance is not explained by the factors in the solution.

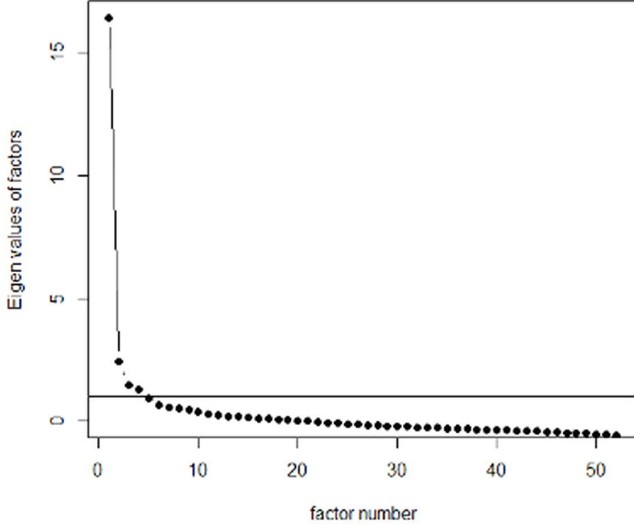

**Fig 1. Scree plot.**

**Table 2. Results of the exploratory factor analysis.**

**Factor Loadings (hide loadings below 0.5)**

| Item | Factor 1 | 2 | 3 | Uniqueness |
|---|---|---|---|---|
| | **1** | **2** | **3** | **Uniqueness** |
| ETQ28 | 0.718 | | | 0.514 |
| ETQ35 | 0.680 | | | 0.424 |
| ETQ1 | 0.670 | | | 0.518 |
| ETQ10 | 0.651 | | | 0.696 |
| ETQ20 | 0.645 | | | 0.655 |
| ETQ46 | 0.626 | | | 0.513 |
| ETQ16 | 0.609 | | | 0.547 |
| ETQ33 | 0.608 | | | 0.694 |
| ETQ22 | 0.566 | | | 0.470 |
| ETQ7 | 0.562 | | | 0.607 |
| ETQ39 | 0.546 | | | 0.512 |
| ETQ49 | 0.546 | | | 0.440 |
| ETQ41* | 0.536 | | | 0.548 |
| ETQ9 | 0.525 | | | 0.616 |
| ETQ2 | 0.511 | | | 0.556 |
| ETQ31 | 0.510 | | | 0.636 |
| ETQ51 | 0.500 | | | 0.413 |
| ETQ14 | | | | 0.811 |
| ETQ44 | | | | 0.706 |
| ETQ3* | | | | 0.589 |
| ETQ52 | | | | 0.662 |
| ETQ15 | | | | 0.782 |
| ETQ19 | | | | 0.708 |
| ETQ26 | | | | 0.336 |
| ETQ25 | | | | 0.599 |
| ETQ21 | | | | 0.706 |
| ETQ34 | | | | 0.544 |
| ETQ13* | | | | 0.874 |
| ETQ11 | | | | 0.841 |
| ETQ23 | | 0.826 | | 0.295 |
| ETQ37 | | 0.739 | | 0.311 |
| ETQ32* | | 0.737 | | 0.406 |
| ETQ4 | | 0.731 | | 0.432 |
| ETQ47* | | 0.594 | | 0.513 |
| ETQ50 | | 0.503 | | 0.573 |
| ETQ42 | | | | 0.642 |
| ETQ36 | | | | 0.477 |
| ETQ45 | | | | 0.642 |
| ETQ5* | | | 0.564 | 0.646 |
| ETQ43* | | | 0.536 | 0.662 |
| ETQ12* | | | 0.530 | 0.629 |
| ETQ18* | | | 0.523 | 0.675 |
| ETQ29* | | | 0.513 | 0.687 |

*(Continued)*

**Table 2.** (Continued)

**Factor Loadings (hide loadings below 0.5)**

| Item | Factor 1 | 2 | 3 | Uniqueness |
|---|---|---|---|---|
| ETQ40* | | | 0.508 | 0.582 |
| ETQ6* | | | | 0.766 |
| ETQ8* | | | | 0.456 |
| ETQ48* | | | | 0.617 |
| ETQ17* | | | | 0.650 |
| ETQ38* | | | | 0.842 |
| ETQ27* | | | | 0.798 |
| ETQ24* | | | | 0.739 |
| ETQ30* | | | | 0.748 |

*Note*. The 'minimum residual' extraction method was used in combination with an 'oblimin' rotation

## Determining factors

The first factor encompassed items from various categories associated with trust. All items of the second factor pertained to the Generalization of knowledge, a domain identified during the questionnaire's development. The third factor only included only negative items. Based on these patterns, the three factors were named "Trust subscale" (ETT_T), "Generalization subscale" (ETT_G), and "Mistrust subscale" (ETT_M). Items with a factor loadings greater than 0.5 were assigned to each factor. Consequently, 17 items were included in the ETT_T subscale, while 6 items were included in each of the ETT_M and ETT_G subscales. The remaining 23 items, with factor loadings below 0.5, were excluded from the analysis at this stage.

## Confirmatory factor analysis

Confirmatory factor analysis was conducted to evaluate the fit of the proposed three-factor structure. Additionally, an effort was made to reduce the number of items in the Trust subscale to 6, aligning it with the item count of the other two subscales. To achieve this, items with the lowest factor loadings were systematically removed. A correlation between residuals for opposite items ("I trust this teacher" from the Trust subscale and "I do not trust this teacher" from the Mistrust subscale) was introduced to refine the model. Table 3 presents the fit indices for the resulting models.

According to model fit indices, the shortened model with three factors fits the data best. Factor loadings and model correlations between factors are shown in Fig 2.

The questions included in the obtained subscales are presented in Table 4.

Cronbach's α for each subscale was =.86,.75,.88.

**Table 3. Model fit indices.**

| Model | $\chi^2$/df | CFI & TLI | RMSEA (90% CI) | SRMR |
|---|---|---|---|---|
| Unidimensional with 29 items | 1057/ 376 = 2.8 | 0.881 & 0.871 | 0.090 (0.84-0.97) | 0.101 |
| Three-factor with 17 items in Trust | 748/ 373 = 2 | 0.934 & 0.929 | 0.067 (0.06-0.074) | 0.076 |
| Three –factor with 6 items in Trust | 243/ 131 = 1.9 | 0.970 & 0.965 | 0.062 (0.05-0.074) | 0.059 |

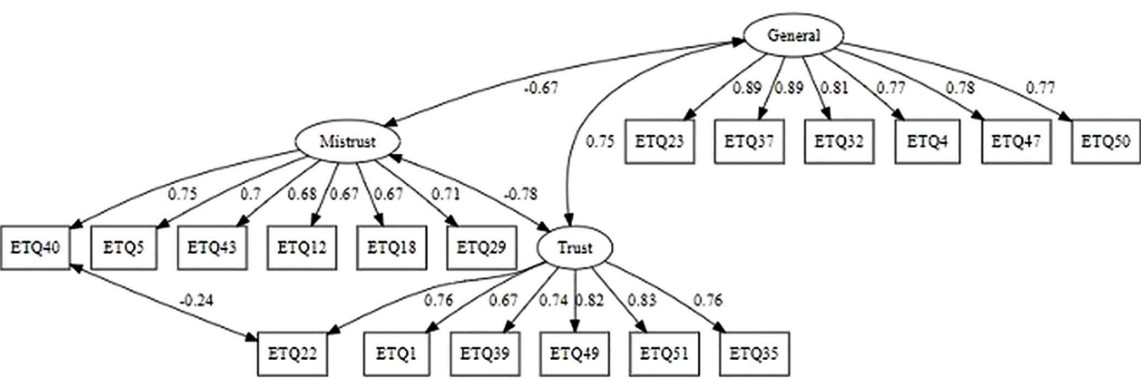

**Fig 2. Results of CFA.**

**Table 4. The composition of the obtained subscale.**

| Trust subscale | Mistrust subscale | Generalization subscale |
|---|---|---|
| ETQ35: Sometimes we laugh and joke together with this teacher | ETQ5: This teacher does not pay attention to me. | ETQ23: Things I learn from this teacher are useful in real life. |
| ETQ1: This teacher often smiles at me. | ETQ43: This teacher never calls me by my name. | ETQ37: Some things I learned from this teacher were useful outside the class. |
| ETQ22: I trust this teacher. | ETQ12: I often feel this teacher does not listen to me. | ETQ32*: I don't see how to apply what we learn in this teacher's class outside of school. |
| ETQ39: This teacher notices when I don't understand something. | ETQ18: This teacher is not interested in what I have to say. | ETQ4: What I learn from this teacher is going to be useful in other settings. |
| ETQ49: This teacher understands me. | ETQ29: This teacher often does not say hello to me, even when they see me. | ETQ47*: I feel what I am learning from this teacher is useless. |
| ETQ51: I feel that my success is important to this teacher. | ETQ40: I do not trust this teacher. | ETQ50: I often find things I learned from this teacher useful in other subjects. |

## Correlations with demographic features

The associations between the ETT subscales and the demographic features of the sample (as presented in Table 1) were examined using Spearman correlation. As hypothesized, none of the three ETT subscales showed significant relationships with demographic characteristics, with one notable exception. Familiarity with the teacher (i.e., knowing the teacher for more than a year) was significantly positively correlated with both the ETT_T scale (r = 0.11, p < 0.01) and the ETT_G scale (r = 0.25, p < 0.01), while showing a significant negative correlation with the ETT_M scale (r = −0.15, p < 0.05). These findings suggest that trust towards the teacher and the generalizability of knowledge increase, and mistrust decreases, as students become more familiar with the teacher over time.

## Correlations with related constructs

The relationships between the ETT subscales and other measures related to the quality of teacher-student relationships in school were explored using Spearman correlation (Table 5). The ETT scales were, as anticipated, related to other measures of student-teacher relationships. Consistent with expectations, ETT_T and ETT_G showed strong positive correlations with positive aspects of the teacher-student relationship, including TSRI subscales for Satisfaction and Instrumental Help, as well as with all aspects of the Working Alliance Inventory (WAI) and positive non-verbal immediacy.

**Table 5. Correlation between the obtained subscales and other measures.**

| Other measures | Trust subscale | Mistrust subscale | Generalization subscale |
|---|---|---|---|
| WAI – Goal | 0.609** | −0.342** | 0.457** |
| WAI – Task | 0.522** | −0.384** | 0.524** |
| WAI – Bond | 0.575** | −0.270** | 0.430** |
| WAI – Total | 0.636** | −0.354** | 0.519** |
| NVI – Positive | 0.418** | −0.163* | 0.173** |
| NVI – Negative | −0.266** | 0.278** | −0.280** |
| TSRI – Satisfaction | 0.772** | −0.549** | 0.578** |
| TSRI – Instrumental help | 0.578** | −0.327** | 0.355** |
| TSRI – Conflict | −0.596** | 0.541** | −0.523** |
| **significant at.01 | | | |
| *significant at.05 | | | |

Additionally, ETT_T and ETT_G were negatively correlated with negative non-verbal immediacy and with conflict within the teacher-student relationship. Conversely, the ETT_M displayed a reverse pattern of associations. Overall, these findings confirm the theoretical assumptions regarding the relationships between the ETT subscales and other constructs, with significant correlations in the expected directions.

### Correlations with learning motivation

The relationships between the ETT subscales and measures of motivation were also examined using Spearman correlation (Table 6). As expected, the ETT showed significant correlations with Student Motivation. Both the ETT_T and ETT_G subscales were significantly positively associated with student motivation, with a weaker association observed specifically for motivation related to studying at a particular school, and no significant association found with the single item measuring general motivation for studying. In contrast, an opposing negative trend of associations was observed for the ETT_M scale.

### Discussion and conclusions

The primary objective of this study was to develop and validate a self-report measure of ET within teacher-student relationships. The results of this study seem to provide preliminary support for the reliability and validity of the ETT. The three factors identified through EFA — Trust, Generalization, and Mistrust — are conceptually consistent and are in line with those in other ET measures [41], while also incorporating the important factor of generalization which is of particularly relevant for educational research [23–25]. Moreover, the internal consistency of the ETT factors was satisfactory to excellent, enabling for a relatively brief yet effective 18-item measure.

**Table 6. Correlation between the obtained subscale and measures of motivation.**

| | Trust subscale | Mistrust subscale | Generalization subscale |
|---|---|---|---|
| Student Motivation | 0.365** | −0.305** | 0.342** |
| Motivation of studying at a particular school | 0.166* | −0.225** | 0.207** |
| Motivation of studying at all | 0.110 | −0.268** | 0.223** |
| **significant at.01 | | | |
| *significant at.05 | | | |

The associations with the conceptually related measures used in both educational and clinical research were consistent with expectations. None of these associations however were high enough to suggest a lack of discriminant validity. Specifically, ETT_T and ETT_G were positively correlated with the working alliance, as well as with the satisfaction and helpfulness in teacher-student relationships and negatively correlated with conflict in the teacher-student relationship. In contrast, associations with the ETT_M scale showed an opposing trend. A similar pattern was obsereved in the association with positive non-verbal immediacy and negative non-verbal immediacy.

The positive associations of the ETT_T and ETT_G, and the negative associations of the ETT_M, with learning motivation align with prior research highlighting the pivotal role of relational factors in study motivation [48,49]. Motivation is, in turn, is one of the strongest predictors of both academic performance and student well-being [50], further supporting the predictive validity of the ETT.

Together, these findings indicate that the ETT may be a promising marker of typical characteristics associated with epistemic trust, although further longitudinal research is required to substantiate these conclusions.

The findings of this study should be interpreted with consideration of several limitations. Firstly, ET is a complex and multifaceted concept that may not be fully captured by self-report measures alone. Consequently, the results of this study should be approached with caution. Further qualitative research, employing observational and ethnographic methods, could help clarify some behavioral aspects of ET and enhance the validity of the results of this study. Self-report was chosen for this study due to the significance of capturing each student's unique, subjective experience, and this instrument has already been proven to be a valuable method for measuring students' cognitive engagement [51–53].

Further evidence for the validity of the ETT could come from a broader socio-economic representation among the participants, as the current sample predominantly comprises students from upper-middle class backgrounds. This homogenous socioeconomic and geographic profile of a participant's sample may limit the applicability of the findings to students from lower-income backgrounds and/or rural contexts, where teacher-student dynamics, nature of the relationships, and access to educational resources may differ significantly. Cultural and social norms around authority, trust, and communication also vary, which may influence how epistemic trust is formed and expressed. As such, the ETT scale may reflect context-specific patterns of trust that may not be universally applicable.

More broadly, the generalizability of the findings is constrained by the limited demographic diversity of the sample. The scale was tested with students from a narrow age range (11–15 years).

Consequently, future studies should expand the age range of participants to explore ET across a wider demographic, including younger school students and older university students. However, adaptations to certain items may be necessary for different age groups. Another limitation for generalizability of this study lies in the fact that it was conducted within a single educational system, which may not capture the full spectrum of student-teacher relational dynamics across different cultural or institutional settings. Future studies should examine the scale's validity and reliability in more diverse populations, including varied socioeconomic, regional, ethnic, and cultural groups, as well as across developmental stages. This would help to ensure the broader applicability and robustness of the ETT measure.

Since data collection for this study, an additional ET measure for clinical contexts — the Epistemic Trust, Mistrust, and Credulity Questionnaire (ETMCQ) [41] — has been published. It may be of interest to investigate the association between the ETMSQ and the ETT within the educational settings. Another important direction for future research is to establish direct links between ET and positive academic outcomes.

Despite these limitations, the present study provides encouraging evidence for the reliability and validity of the ETT as a valuable tool for use with school student populations. The final 18-item version demonstrates strong psychometric properties, capturing varying levels of epistemic trust, mistrust, and generalization of learned knowledge within student-teacher relationships.

Beyond the psychometric validation, the "Generalization" subscale has important theoretical and practical implications. Conceptually, this subscale captures students' subjective sense that knowledge acquired from a specific teacher is

meaningful and transferable beyond the immediate classroom context — a key feature of epistemic trust. In educational terms, this reflects not only the credibility of the teacher but the perceived authenticity and relevance of the learning process.

From a practical perspective, high scores on this subscale suggest that students view their learning as useful and applicable in real life — a recognized driver of both engagement and long-term retention.

Interventions aimed at improving students' epistemic trust could incorporate training teachers in ostensive communication (e.g., using name recognition, responsiveness, humor) as well as pedagogical strategies that support knowledge transfer and relevance.

At the policy level, the ETT could inform teacher development programs or be used as a diagnostic tool to identify gaps in classroom relational climates.

Further research across diverse demographics could enhance our understanding of the role of ET in education, shedding light on the impact of epistemic trust on academic achievement.

## Supporting information

**S1 Questionnaire. The epistemic trust towards teacher questionnaire.**
(DOCX)

**S3 Checklist. Human participants research checklis.**
(DOCX)

**S4 Questionnaire. Inclusivity in global research.**
(DOCX)

## Author contributions

**Data curation:** Sergei Tarasov.

**Formal analysis:** Alex Desatnik, Sergei Tarasov.

**Investigation:** Alex Desatnik.

**Methodology:** Alex Desatnik, Peter Fonagy.

**Project administration:** Alex Desatnik.

**Resources:** Maxim Yakubovskiy.

**Software:** Sergei Tarasov.

**Supervision:** Peter Fonagy.

**Writing – original draft:** Alex Desatnik, Maxim Yakubovskiy, Sergei Tarasov.

**Writing – review & editing:** Alex Desatnik, Maxim Yakubovskiy.

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
