## [Decision Letter · Decision Letter 0]

4 Jul 2025

Dear Dr. Yakubovskiy,

We look forward to receiving your revised manuscript.

Kind regards,

Amir Karimi, PhD

Academic Editor

PLOS ONE

Journal Requirements:

3. In the online submission form, you indicated that “Our study’s minimal underlying data are within the manuscript, and all other data can be uploaded to a repository upon request”.

5. Please remove all personal information, ensure that the data shared are in accordance with participant consent, and re-upload a fully anonymized data set.

Additional guidance on preparing raw data for publication can be found in our Data Policy (https://journals.plos.org/plosone/s/data-availability#loc-human-research-participant-data-and-other-sensitive-data) and in the following article: http://www.bmj.com/content/340/bmj.c181.long .

Reviewers' comments:

Reviewer's Responses to Questions

**Comments to the Author**

1. Is the manuscript technically sound, and do the data support the conclusions?

Reviewer #1: Yes

Reviewer #2: Yes

2. Has the statistical analysis been performed appropriately and rigorously?

Reviewer #1: Yes

Reviewer #2: Yes

3. Have the authors made all data underlying the findings in their manuscript fully available?

Reviewer #1: Yes

Reviewer #2: Yes

4. Is the manuscript presented in an intelligible fashion and written in standard English?

Reviewer #1: Yes

Reviewer #2: Yes

Reviewer #1: Dear Editor,

Thank you very much for inviting me to review this study. I believe that the manuscript could be accepted with minor revisions in terms of grammar and some APA issues. I got familiarized with epistemic trust a few months ago and wondered it would be a good idea to consider its implications in education. I am very happy to see that the esteemed authors conducted this study. The authors have skillfully provided a very thorough account of the literature review. In addition, the study has a well-justified design, and the data have been analyzed accurately. Meanwhile, the discussion section is very convincing. Overall, this is a high-quality manuscript which could be accepted for publication.

Last but not the least, it would be very necessary to provide more information about the implications of epistemic trust in education. For example, the authors may refer to more potentially-related correlates of epistemic trust in learning.

Best regards,

Reviewer

Reviewer #2: Abstract:

The abstract introduces a promising and timely study on the development and validation of the Epistemic Trust Towards Teacher (ETT) scale. However, several improvements are recommended to enhance its clarity, precision, and alignment with reporting standards for psychometric research:

• The results section of the abstract should briefly include model fit indices from the confirmatory factor analysis (e.g., CFI, RMSEA) and reliability coefficients (e.g., Cronbach’s alpha).

• Replacing phrases like “relatively distinct and theoretically coherent” with more precise statements (e.g., “the three factors showed good discriminant validity and aligned with theoretical expectations”) would improve clarity.

Introduction:

The introduction offers a strong and well-referenced theoretical foundation for the study, effectively bridging epistemic trust from its developmental and evolutionary psychology origins to its relevance in educational contexts. The justification for creating a context-specific instrument tailored to the student–teacher relationship is compelling and timely. However, the section would benefit from streamlining and greater clarity. Some ideas particularly those related to the evolutionary basis of learning and the role of generalization are repeated and could be condensed to improve readability. Additionally, the inclusion of methodological details (e.g., questionnaire development and hypotheses) in the introduction disrupts the logical flow and may be better placed in the Methods section. Improving transitions between themes and refining some sentences for grammar and conciseness will further enhance the narrative clarity and strengthen the impact of this important contribution.

Methods

The Methods section presents a solid framework for instrument development, with commendable efforts taken to ensure theoretical grounding, expert review, and psychometric rigor. The initial item pool was thoughtfully developed across six conceptual dimensions, and the multi-phase validation design—comprising EFA, CFA, and correlation-based validity testing—is appropriate and aligned with current best practices in scale development. The use of multiple well-validated external measures (WAI, NVI, TSRI, SMS) to assess convergent and divergent validity is particularly noteworthy and strengthens the methodological integrity of the study.

However, several aspects of the section would benefit from clarification and minor revisions:

Clarity and Structure:

The section is densely written and would be easier to follow with clearer subheadings (e.g., "Item Development," "Participants and Procedures," "Validation Instruments," "Statistical Analysis").

Sentences such as those in lines 185–193 are long and would benefit from being split and simplified for clarity.

Item Reduction Procedure:

While the expert review process is mentioned, the criteria for item rejection (e.g., item-total correlations, theoretical redundancy) are not fully elaborated.

Please clarify how final items for CFA were selected beyond expert input and whether inter-rater agreement among experts was assessed.

Sampling and Demographics:

The demographic table is informative, but the sample is highly homogeneous (urban, mostly upper-middle class). This limitation should be acknowledged either here or in the Discussion section as it affects generalizability.

Ethical Procedures:

The ethical approvals and consent procedures are adequately described. It might help to explicitly state that data were anonymous and voluntary to reinforce adherence to ethical standards.

discussion

The discussion section presents a clear and thoughtful interpretation of findings, effectively connecting empirical results to the theoretical constructs introduced earlier in the manuscript. The authors appropriately highlight the study’s main contributions, including the validation of a psychometrically sound, theoretically grounded, and practically relevant measure of epistemic trust in student–teacher relationships. The inclusion of generalization as a distinct factor adds novel value to the field of educational psychology.

Nevertheless, several improvements are recommended to enhance clarity, precision, and scientific rigor:

Overuse of General Statements

Phrases such as “seem to provide preliminary support” and “may be a promising marker” occur frequently and can be more assertively stated, especially when the evidence is clear and the statistical results are strong. I suggest to rephrase to reflect confidence where justified by the data (e.g., “provide preliminary evidence” or “indicate potential”).

Limited Depth in Interpretation

While the findings are restated clearly, the discussion would benefit from deeper theoretical interpretation. For example, how might the "generalization" subscale inform classroom practices or intervention design? I suggest to expand on practical implications for teachers or policy recommendations based on ETT findings.

Acknowledgment of Limitations

The limitations are appropriately acknowledged (e.g., use of self-report, socioeconomic homogeneity), though the discussion of sampling limitations could be more explicitly tied to generalizability. I suggest to clarify how the upper-middle-class, urban sample may limit cross-cultural or lower-income applications.

**Do you want your identity to be public for this peer review?** For information about this choice, including consent withdrawal, please see our Privacy Policy

Reviewer #1: No

Reviewer #2: No

---

## [Author Response · Author response to Decision Letter 1]

27 Jul 2025

Dear Dr. Karimi and Reviewers,

We would like to thank you and the reviewers for the thoughtful and constructive feedback on our manuscript, titled “Epistemic trust towards teacher questionnaire: Development and preliminary validation.” We greatly appreciate the opportunity to revise our submission and are pleased that the reviewers found the manuscript to be of high quality, with a well-justified design, accurate data analysis, and a convincing discussion. We are particularly grateful for the reviewer’s encouraging comments regarding the theoretical potential of epistemic trust in educational contexts. We are also thankful for the reviewers’ recognition of the study’s theoretical grounding, methodological rigor, and practical relevance.

Below, we provide point-by-point responses to all comments, describing the revisions made in response.

Reviewer #1 Comment:

“It would be very necessary to provide more information about the implications of epistemic trust in education. For example, the authors may refer to more potentially-related correlates of epistemic trust in learning.”

Response:

We thank the reviewer for this thoughtful and constructive suggestion. In response, we have expanded the Discussion section to offer a deeper theoretical interpretation of the “Generalization” subscale and its relevance for educational practice. We highlight how this dimension reflects students’ perception of the real-world applicability of knowledge and propose implications for teacher communication strategies, classroom design, and teacher training. We also briefly discuss how this measure can be used to guide interventions or inform educational policy. This addition is now included on page 30 of the revised manuscript.

Additionally, we have addressed the minor revisions suggested regarding grammar and APA style formatting to ensure clarity and adherence to publication standards.

Reviewer #2 Comment (Abstract):

“The abstract should include model fit indices from the CFA and reliability coefficients. Additionally, phrases such as “relatively distinct and theoretically coherent” should be replaced with more precise language.”

Response:

We thank the reviewer for these insightful suggestions. We have revised the abstract to enhance clarity and align it with psychometric reporting standards. Specifically, we now report CFA model fit indices (χ²/df, CFI, TLI, RMSEA) and Cronbach’s alpha coefficients for each subscale. We have also replaced vague phrasing with more precise language describing the discriminant validity and theoretical alignment of the three factors. The revised abstract appears on page 2 of the manuscript.

Reviewer #2 Comment (Introduction):

“While the introduction is strong, some theoretical content (particularly on the evolutionary basis of learning and generalization) is repeated and could be condensed. Methodological content (e.g., item development and hypotheses) may be better suited for the Methods section. Transitions and grammar should also be improved.”

Response:

We appreciate this valuable feedback. The introduction has been revised to improve readability and logical flow. We have removed repetitive content, streamlined the discussion of evolutionary theory and generalization, and relocated methodological details (e.g., scale development procedures, hypotheses) to the Methods section on pages 7-8 of the Manuscript. We have also refined transitions and edited for clarity and conciseness throughout the section.

Reviewer #2 Comment (Methods – Clarity and Structure):

“The Methods section is densely written and would benefit from subheadings (e.g., “Item Development,” “Participants and Procedures,” etc.). Long sentences (e.g., lines 185–193) should be simplified.”

Response:

Thank you for this constructive suggestion. We have restructured the Methods section to include clear subheadings: “Development of the epistemic trust towards teachers questionnaire,” “Study hypothesis,” “The development of the scale,” “Participants and Procedures,” and “Other Measures.” We have also revised lengthy sentences for clarity and readability, as advised.

Reviewer #2 Comment (Methods – Item Reduction Procedure):

“The criteria for item rejection are not fully explained. Please clarify how items were selected for CFA beyond expert input and whether inter-rater agreement was assessed.”

Response:

We thank the reviewer for pointing this out. We have expanded the manuscript to clarify our item reduction procedure. Specifically, items were evaluated by seven international experts based on clarity, relevance, and redundancy. Items were retained if at least five experts rated them as highly relevant and non-redundant. While we did not compute formal inter-rater agreement statistics, a structured consensus procedure modeled on the Delphi method was used, incorporating independent review and iterative feedback. These details have been added to the “The development of the scale” section on page 10.

Reviewer #2 Comment (Sampling and Demographics):

“The sample is highly homogeneous (urban, upper-middle-class). This limitation should be acknowledged, as it affects generalizability.”

Response:

We fully agree with this critical point. We have revised the Limitations section (pages 28–30) to explicitly acknowledge the socioeconomic and geographic homogeneity of our sample. We discuss how this limitation affects the generalizability and interpretation of findings across more diverse or cross-cultural populations. We also note the need for future validation of the scale in broader and more diverse samples.

Reviewer #2 Comment (Ethical Procedures):

“The ethical procedures are adequately described, but it would help to state explicitly that data were anonymous and voluntary.”

Response:

We appreciate this suggestion. We have now explicitly stated that participation was both anonymous and voluntary, in order to reinforce adherence to ethical standards. This clarification appears on pages 10–11 of the revised manuscript.

Reviewer #2 Comment (Discussion – Overuse of General Statements):

“Phrases such as “seem to provide preliminary support” or “may be promising” are overused. Please use more assertive language where justified by strong data.”

Response:

Thank you for this helpful observation. We have revised the Discussion section to reduce hedging language and adopt more confident phrasing where the data are statistically robust. For example, we now state that the findings “a similar pattern was observed,” “these findings indicate,” “study provides encouraging evidence” rather than “seem to provide,” and have made similar adjustments throughout the paper.

Reviewer #2 Comment (Discussion – Limited Depth in Interpretation):

“The discussion would benefit from deeper theoretical interpretation—particularly of the “Generalization” subscale—and its implications for educational practice and policy.”

Response:

We thank the reviewer for this important recommendation. We have expanded the Discussion section to offer a deeper interpretation of the “Generalization” subscale and its practical relevance. We propose that this dimension reflects students’ perception of the real-world applicability of knowledge, which may inform strategies in classroom communication, instructional design, and teacher training. We also comment on how this construct could guide targeted interventions and educational policy. This revision is included on page 30 of the revised manuscript.

Reviewer #2 Comment (Discussion – Acknowledgment of Limitations):

“Limitations are generally acknowledged, but sampling limitations should be more explicitly tied to generalizability.”

Response:

We agree and have revised the Limitations section (pages 28–30) to more clearly connect sample characteristics — especially the upper-middle-class, urban background of participants — to constraints on generalizability. We emphasize the importance of future studies validating the ETT scale in more diverse socioeconomic and cultural settings.

Once again, we are grateful for the opportunity to revise the manuscript and for the reviewers’ positive evaluation of our work. We sincerely thank the Academic Editor and both Reviewers once again for their detailed and insightful comments. These suggestions significantly improved the clarity, depth, and impact of the manuscript.

Sincerely,

Max Yakubovskiy

Department of Educational Policy Studies,

School of Education,

University of Wisconsin-Madison, WI, USA

yakubovskiy@wisc.edu

---

## [Editor Report · Decision Letter 1]

15 Aug 2025

Epistemic trust towards teacher questionnaire: Development and preliminary validation

PONE-D-24-44350R1

Dear Dr. Yakubovskiy,

We’re pleased to inform you that your manuscript has been judged scientifically suitable for publication and will be formally accepted for publication once it meets all outstanding technical requirements.

Kind regards,

Amir Karimi, PhD

Academic Editor

PLOS ONE
---

## [Editor Report · Acceptance letter]

PONE-D-24-44350R1

PLOS ONE

Dear Dr. Yakubovskiy,

I'm pleased to inform you that your manuscript has been deemed suitable for publication in PLOS ONE. Congratulations! Your manuscript is now being handed over to our production team.

Kind regards,

on behalf of

Dr. Amir Karimi

Academic Editor

PLOS ONE